# Prevalence of depression and associated factors among obstetric care providers at public health facilities in the West Arsi Zone, Ethiopia: Cross-sectional study

**Solomon Seyife Alemu**[1]*, **Mohammedamin Hajure Jarso**[1], **Zakir Abdu Adem**[2], **Gebremeskel Mulatu Tesfaye**[2], **Yadeta Alemayehu Workneh**[2], **Wubishet Gezimu**[2], **Mustefa Adem Hussen**[2], **Aman Dule Gemeda**[2], **Sheleme Mengistu Teferi**[1], **Lema Fikadu Wedajo**[3]

1 Department of Midwifery and Psychiatry, College of Health Sciences, Madda Walabu University, Sheshemene, Ethiopia, 2 Department of Psychiatry, Nursing, and Midwifery, College of Health Science, Mattu University, Mattu, Ethiopia, 3 Department of Midwifery, Institute of Health Sciences, Wollaga University, Nekemte, Ethiopia

* soleseifa@gmail.com

**Data Availability Statement:** Regarding issues of data availability; the current manuscript is a part of an ongoing research project owned by Madda

## Abstract

### Background

Depression is a severe and treatable mental illness that significantly affects individuals' daily activities. Obstetric care providers are the most vulnerable group for depression because they work in an emergency to save two lives at a time, share the stress of women during labor, and are at great risk for contamination.

### Objectives

To assess depression and associated factors among obstetric care providers working in public health facilities.

### Method and materials

A cross-sectional study was conducted among 423 obstetric care providers working in public health facilities found in the West Arsi Zone, Ethiopia, from June 1 to 30, 2023. Study participants were selected through a simple random sampling technique. A pretested, face-to-face interviewer-administered structured questionnaire was used to collect data. Bi-variable and multivariable logistic regression analyses were employed to identify factors associated with depression. The level of statistical significance was declared at P < 0.05 with a 95% CI.

### Conclusions and results

Overall, the prevalence of depression among obstetric care providers was 31.1% (95% CI: 26.6%, 35.5%). Marital status not in union (AOR = 2.86, 95%CI: 1.66, 4.94), working more than 40 hours per week (AOR = 2.21, 95%CI: 1.23, 3.75), current substance use (AOR = 2.73, 95%CI: 1.64, 4.56), not being satisfied with their job (AOR = 3.52, 95%CI: 2.05, 6.07)

Walabu University Shashemenne Campus. All relevant data are included within the paper. Full data is guarded carefully by our research team for the purpose of this ongoing scientific study. Interested, qualified researchers can request data access by contacting the following research coordinators: 1. Habtemu Jarso (BSc, MPH-E, Assis. Professor) Shashemene Campus, Madda Walabu University Institutional email: habtemu.jarso@mwu.edu.et 2. Solomon Seyife (Corresponding author) email: SoleSeifa@gmail.com.

**Funding:** The author(s) received no specific funding for this work.

**Competing interests:** The authors have declared that no competing interests exist.

**Abbreviations:** AOR, Adjusted Odds Ratio; COR, : Crude Odds Ratio; CI, Confidence Interval; GP, General Practitioners; HCW, Health Care Workers; IESO, Integrated Emergency Surgery Officers; OSLO, Oslo Social Support Scale; SPSS, Statistical Package for Social Sciences; WHO, World Health Organization.

and having burnout symptoms (AOR = 5.11, 95%CI: 2.95, 8.83) were factors significantly associated with depression.

## Recommendations

We recommend that health professionals take care of themselves and avoid substance use. We also recommended that stakeholders enhance job satisfaction and avoid burnout by implementing various programs, like raising wages for workers, increasing staff members, offering various benefits, and regularly monitoring issues that arise.

## Introduction

Depression is a severe and treatable mental illness that negatively affects an individual's feelings, the way he or she thinks, and how they act. If it can't be managed early, it can lead to a variety of emotional and physical problems and increase disability to function at the workplace and home [1]. Depressive disorder is yet an underappreciated mental disorder in the population, which is manifested by depressed symptoms including mood or loss of pleasure or interest in any activities that one's individual performs in his or her daily life [2–4].

Depression is quite different from ordinary mood changes or a feeling that happens to an individual due to his or her daily life situation [5]. Instead, it can affect all aspects of an individual's life, including relationships with family, friends, and society. Depression can happen to anyone; exceptionally, people who have a previous or current history of abuse, severe losses, or other stressful events are more likely to develop depression [6]. There are a lot of causes of depression among them job-related factors such as emergency activities, work hours, workload, challenges with daily activities, relationship with colleagues, and access to paid leave impact the well-being of workers, their families, and their communities [7].

Worldwide healthcare workers (HCWs) are classified among the high-risk groups to develop mental health disorders [8]. More specifically, studies have shown that they experience high levels of psychological distress such as depression, anxiety, and poor sleep quality due to their daily activities related to their profession [9, 10]. The common reason that healthcare providers have these disorders is that most of the time they perform their job in restlessness, anger, frustration, insomnia, and fatigue situations [9].

Globally, the World Health Organization (WHO) report revealed that nearly 280 million general populations suffer from depression [11], which might include healthcare providers. An umbrella review that included 110 studies conducted during the COVID-19 pandemic showed that 24.83% of healthcare providers suffered from depression [12]. In high-income countries, depression among healthcare providers ranged from 21.53% to 32.77% [10, 13]. In Africa, studies revealed the largest magnitude of depression among healthcare providers, which ranges from 27.8% to 54.5% [14, 15].

Almost all organ systems are affected by depression, including the cardiovascular, kidney, gastrointestinal, nervous, and immune systems [16]. Depression is also affecting social interactions, which are the building blocks of interpersonal social networks for good functional social relationships. Individuals with depressive symptoms have a greater chance of being isolated in their social relationships, which can further increase their symptoms [17]. Furthermore, depression has an economic burden; for instance, in the United States, it cost an estimated $236 billion in 2018, an increase of more than 35% since the 2010 cost of depression disorder [18].

There are identified factors that cause depression among healthcare providers. The studies revealed that job-related factors such as marital status, sex of study participants, working hours, workload, job satisfaction, and access to paid leave were factors that led to depression among healthcare providers, which affects the well-being of workers, their families, and their communities [7, 12, 14, 19].

By 2030, depression symptoms are predicted to overtake other causes as the second largest contributor to disability-adjusted life years (DALYs) [20]. To overcome this problem and other mental disorders, reports from the Compressive Mental Health Action Plan 2030 aim to improve mental health by strengthening effective leadership, integrated and responsive community-based care, evidence, and research [21]. World Health Organization transforms this to mental health for all and calls on all countries to accelerate implementation of this plan [22]. However, little is known about depression among obstetric care providers in Africa, including Ethiopia.

Obstetrics care providers are highly at risk for mental health problems like depression as a result of stress, trauma, grief, and shortages of resources they face in their working environment due to the nature of the environment of their work. Therefore, this study aimed to assess the prevalence of depression and associated factors among obstetric care providers in the West Arsi Zone in 2023.

## Methodology

### Study area, study design and period

The cross-sectional study was conducted among obstetrics care providers working in public health facilities found in the West Arsi zone from June 1 to 30, 2023. This zone is found in the Oromia region of southern Ethiopia. The zone has one referral hospital, two general hospitals, four primary hospitals, and 85 health centers. The data obtained from the zonal health office revealed that there were 5 gynecologists, 65 general practitioners (GPs), 567 midwives, 17 integrated emergency surgical officers (IESO), 188 nurses, and 178 health officers serving maternal and child health in those health facilities. The zone has 647,690 women in the reproductive age group and 101,558 estimated deliveries annually in 2023.

**Source population.**    All obstetrics care providers working in public health facilities found in West Arsi zone.

**Study population.**    A selected obstetric caregivers who were working in selected public health facilities in West Arsi Zone and found during data collection.

### Eligibility criteria

Obstetrics care providers who have worked for at least six months in public health facilities located in the West Arsi zone were included in the study, whereas individuals who were on maternity leave as well as on annual leave were excluded.

### Sample size determination

Sample size was calculated using a single population proportion formula by considering the following assumptions: a proportion (p) of 50%, a Z value at a 95% confidence level (Z$\alpha$/2) = 1.96, and a 5% degree of precision by using $n = Z^2*(1-P)/d^2$. By substituting the in the formula, the minimum required sample size was 384 obtained. Adding a 10% non-response rate, the final sample size 423 taken. Where; n = the required sample size; z = the value of the standard normal curve score corresponding to the given confidence level = 1.96; p = proportion of population which is 50%; d = the permissible margin of error is 5%.

## Sampling technique and procedure

The study was conducted among obstetric care providers found in West Arsi public health facilities. According to the zonal health bureau information, West Arsi Zone has a total of 92 public health facilities, with 1020 health care providers working in the obstetrics unit. By following the recommendations of the WHO on sampling techniques, 40% of health facilities were included in the study. So, from the total of 92 public health facilities found in the zone, thirty-seven health facilities were selected by using the lottery method. So three public hospitals namely Malka Oda General Hospsiital, Arsi Nagelle Primary Hospital, Shashashemenne Compressive Referral Hospital as well as other selected thirty four public health center found in the zone was included. The total number of healthcare providers working at an obstetric unit in selected public health facilities, which was obtained from the zonal health bureau, was 708. From similar data obtained from zonal health office those obstetric care providers were that serving selected public health facilities categorized as 456 Midwifery professional, 108 Nursing, 13 IESO, 39 General practitioners, 89 Health officers and 3 Obstetrician. Before the selection of study participants, obstetric care providers were proportionate allocations to each health facility were made. Finally, from the list, the required sample sizes of 423 obstetric care providers were selected by simple random sampling techniques.

## Study variables

**Dependent variable.** Depression.

**Independent variables.** *Socio-demographic variables*. Age, marital status, educational status, residence religion, ethnicity, monthly income

*Service, personal, social, and clinical related factors*. Types of health facilities, years of service, Work hours per week, types of health professional, history of mental illness, history of chronic illness, family history of mental illness, current substance uses, job satisfaction, burnout and social support.

## Operational definition

**Depression.** The Patient Health Questionnaire-9, with their responses ranging from "0" (not at all) to "3" (nearly every day), was used for the assessment of depressive symptoms, with a total score of nine items ranging from 0 to 27. Based on the previous study, those Obstetrics care providers who scored $\geq 10$ were considered to have depression symptoms, and a code '1' was given. While those who scored $< 9$ were considered to have no depression symptoms, and a code '0' was given by Patient Health Questionnaire-9 tool [23].

**Job satisfaction.** The job satisfaction was categorized into five Likert scales (from strongly disagree to strongly agree). Those Obstetrics care providers were considered as satisfied with their job if they answered greater than or equal to the mean value 50% [19].

**Burnout.** There are 19 items to assess symptoms of burnout syndrome (19 items) that are categorized into four Likert scales 0 (never), 1 (sometimes), 2 (often), and 3 (always). The sums of the scores of the rating of the items were calculated with a total minimum score of 0 to a maximum score of Healthcare. Obstetrics care providers that scored more than 23 out of 57 were considered to have burnout symptoms [4].

**Social support.** Social support will be. The OSSS-3 sum score then will be operationalized into three broad categories of social support based the score as follows. Oslo Social Support Scale (OSLO) was used, and classified as 3–8 poor social support code '1', 9–11 moderate social support code '2', and 12–14 strong social Support code '3' was given [24].

## Data collection tool and procedure

A structured, interviewer-administered questionnaire was used to collect the data. The questionnaires have questions to assess socio-demographic factors, service-related, personal-related, social-related, and clinical-related factors, as well as questions to assess job satisfaction, burnout, and depression. The questionnaires to assess socio-demographic factors, service-related, personal-related, social-related, and clinical-related factors were developed from previous literature. The tool to assess symptoms of burnout syndrome [4], job satisfaction [19], and social support [24] was validated in Ethiopia.

The dependent variable was assessed by the Patient Health Questionnaire-9, which has nine items with a minimum score of 0 and a maximum score of 27. This tool was validated in the Ethiopian population with a Cronbach's alpha of 0.92 [14, 23]. The questionnaire was prepared in English and later translated into Afan Oromo to check the consistence of questionnaire, and English language was used for data collection. Thirty-seven bachelor's degree data collectors and nine master's degree supervisors were recruited for data collection. Data collection was supervised by supervisors on a daily basis.

## Data quality control

To assure data quality the tool was translated from the English language to Afan Oromo, then retranslated back into English. To assure the quality of the data, a pretest was done in Arsi Zone health facilities on a sample of 10% of the sample size of obstetric care providers. Training was given to both data collectors and supervisors. The collected data was checked for completeness daily by data collectors and supervisors. Finally, data cleaning was done using SPSS 26 version software.

## Data processing and analysis

Data was edited, coded, and entered into Epidata version 3.1 software. Then, the data was exported to Statistical Package for Social Sciences (SPSS) version 26 software for cleaning and further analysis. Descriptive statistics such as mean, standard deviation, and percentage were determined. Some variables like depression, burnout, job satisfaction and social support were dichotomized based on the operational definitions their operational definitions. Then association between the outcome variable and each independent variable was first seen in the binary logistic regression model.

In the second step, independent variables with a p-value $< 0.25$ were retained and entered into the binary logistic regression model for multivariable analysis. The degree of association between outcome and independent variables was determined using the odds ratio with a CI of 95% and p-value. A p-value of $< 0.05$ was considered a cutoff point to declare that there is a statistically significant association between dependent and independent variables. The fitness of the model was tested by Hosmer-Lemeshow goodness-of-fit test, and the value was 0.95. Finally, the result was presented in text, table, graph, and narrative form.

## Ethical consideration

Ethical clearance was obtained from Madda Walabu University, Sheshemene Campus School of Health Science Ethical Review Committee with reference number MWUSH/1426. Written consent was obtained from each study subject before data collection, and the purpose of the study was explained to the respondents. All methods were conducted following the Helsinki Declaration. Further, to protect the confidentiality of the information, names and

identification were not included in the written questionnaires. During the data collection, each study subject was told that their participation would be voluntary.

## Result

### Socio-demographic characteristics of study participants

In this study, 415 obstetric care providers were interviewed, making the response rate 98.1%. Of the total participants, 264 (63.6%) were female. The age of participants ranges from 18 to 52 years, with a mean age of 31.80 (± 4.68 SD), and 253 (61%) were married. Concerning the educational status of study participants, about two-thirds (66%) had a degree (**Table 1**).

### Service, personal, social, and clinical related factors of study participants

From the total study participants, 57.1% of obstetric care providers are serving in health centers and have had five years or fewer services. In addition, 56.9% of obstetric care providers serve more than 40 hours per week. About three-fourths of the obstetric care providers found in this study were midwives. As identified by this finding, 48.5% of study participants had a fear of contamination with a communicable disease. Of all respondents, 42.7% were currently

**Table 1. Socio-demographic characteristics of study participants on prevalence of depression among obstetric care providers in West Arsi Zone, Ethiopia 2023.**

| Variables | Categories | Frequency (415) | Percentage (100%) |
|---|---|---|---|
| Sex of study participants | Male | 151 | 36.4 |
| | Female | 264 | 63.6 |
| Age of study participants | 18–25 | 150 | 36.1 |
| | 26–35 | 102 | 24.6 |
| | 36–44 | 90 | 21.7 |
| | ≥45 | 73 | 17.6 |
| Marital status of study participants | In union | 253 | 61 |
| | Not in union | 162 | 39 |
| Ethnicity | Oromo | 241 | 58.1 |
| | Ahmara | 117 | 28.2 |
| | Tigre | 47 | 11.3 |
| | Others* | 10 | 2.4 |
| Religion | Orthodox | 153 | 36.9 |
| | Muslim | 163 | 39.3 |
| | Protestant | 97 | 23.4 |
| | Others** | 2 | 0.5 |
| Educational level of study participants | Diploma | 83 | 20 |
| | Degree | 274 | 66 |
| | Masters and above | 58 | 14 |
| Residences | Urban | 182 | 43.9 |
| | Rural | 233 | 56.1 |
| Monthly income (in Ethiopian birr) | ≤6,192 | 151 | 36.4 |
| | 6,163–8,016 | 86 | 20.7 |
| | 8,017–10,169 | 129 | 31.1 |
| | ≥10,170 | 49 | 11.8 |

Foote note

* Gurage, Silte and Wolayita

** Catholic and Waqefata

**Table 2. Service, personal, social, and clinical related factors on prevalence of depression among obstetric care providers in West Arsi Zone, Ethiopia 2023.**

| Variables | Categories | Frequency (415) | Percentage (100%) |
|---|---|---|---|
| Types of health facilities | Health center | 237 | 57.1 |
| | Primary hospital | 135 | 32.5 |
| | Referral hospital | 43 | 10.4 |
| Year of service | 1–5 | 237 | 57.1 |
| | 6–10 | 135 | 32.5 |
| | ≥11 | 43 | 10.4 |
| Work hours per week | ≤40Hrs | 179 | 43.1 |
| | >40Hrs | 236 | 56.9 |
| Types of health providers | Midwifery | 311 | 74.9 |
| | Nursing | 63 | 15.2 |
| | IESO | 11 | 2.7 |
| | GP | 10 | 2.4 |
| | Obstetrician | 2 | 0.5 |
| | Health officer | 18 | 4.3 |
| History of mental illness | Yes | 6 | 1.4 |
| | No | 409 | 98.6 |
| History of chronic illness | Yes | 19 | 4.6 |
| | No | 396 | 95.4 |
| Family history of mental illness | Yes | 13 | 3.1 |
| | No | 402 | 96.9 |
| Fear of contamination | Yes | 200 | 48.2 |
| | No | 215 | 51.8 |
| Current substance uses | Yes | 177 | 42.7 |
| | No | 238 | 57.3 |
| Job satisfaction | Satisfied | 164 | 39.5 |
| | Not Satisfied | 251 | 60.5 |
| Burnout | No | 280 | 67.5 |
| | Yes | 135 | 32.5 |
| Social Support | Poor | 172 | 41.4 |
| | Moderate | 195 | 47 |
| | Strong | 48 | 11.6 |

using substances. Around sixty percent of study participants had no satisfaction with their current job, and 32.5% had symptoms of burnout. Furthermore, 47% of the study participants had moderate social support (**Table 2**).

## Prevalence of depression among obstetric care providers in West Arsi Zone, Ethiopia 2023

This study revealed that 31.10% (95% CI: 26.6%, 35.5%) of obstetric care providers had symptoms of depression (**Fig 1**).

## Factors associated with depression

Bivariable logistic regression analysis was done with a 95% CI with COR to identify candidate variables, and AOR was calculated in multivariable logistic regression. The sex of the study participants, marital status, educational status, monthly income, working hours per week, current substance use, social support, job satisfaction, and burnout were transferred to

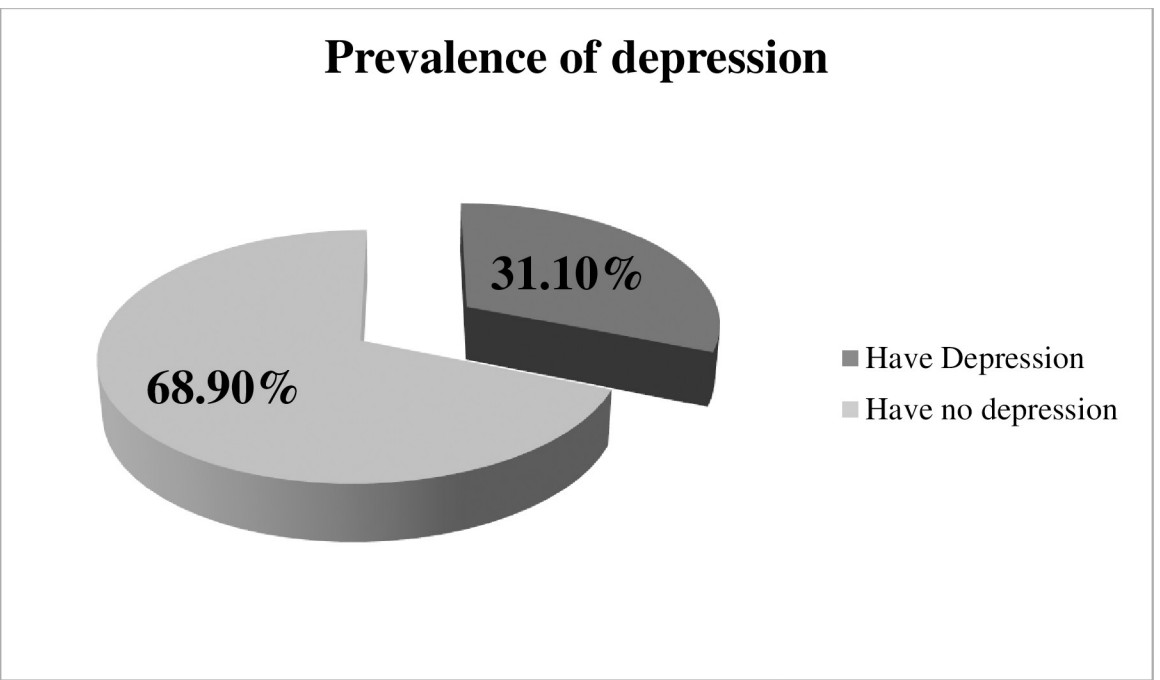

**Fig 1. Prevalence of depression among obstetric care providers.**

multivariable logistic regression. Finally, marital status, working hours per week, current substance use, job satisfaction, and burnout were identified as factors associated with depression (S1 Table).

The odds of having depression were 2.86 times (AOR = 2.86, 95%CI: 1.66, 4.94) higher in those who had no in-union compared to those who had union in marital status. The odds of having depression among obstetric care providers serving more than 40 hours per week were 2.21 times (AOR = 2.21, 95%CI: 1.23, 3.75) higher compared to their counterparts. Current substance use is a factor that is associated with depression. The odds of depression were 2.73 times (AOR = 2.73, 95%CI: 1.64, 4.56) higher among substance users compared to non-users. Job satisfaction and burnout are other factors identified in this study.

The odds of having depression among obstetric care providers that not satisfied with their jobs were 3.52 times (AOR = 3.52, 95%CI: 2.05, 6.07) higher compared to those who were satisfied with their jobs. Lastly, the odds of depression among participants who had burnout symptoms were 5.11 times (AOR = 5.11, 95%CI: 2.95–8.83) higher than those who had no symptoms of burnout (**Table 3**).

## Discussion

This study identified the prevalence of depression and associated factors among obstetric care providers working at public health facilities in the West Arsi Zone, Ethiopia, in 2023. As identified from this study, 31.1% of study participants had depression, and marital status, working hours per week, current substance use, job satisfaction, and burnout were identified as factors significantly associated with depression.

The result of this study is in line with the study conducted in Dessie Compressive Specialized Hospital, Ethiopia [14] which was 27.8%. This might be due to the similarity of setups in the same country. In addition, health providers get similar advantages and careers and are

**Table 3. Binary and multivariable logistic regression on factors associated with depression among obstetric care providers in West Arsi Zone, Ethiopia, 2023.**

| Variables | Categories | Depression | | COR (95% CI) | AOR (95% CI) |
|---|---|---|---|---|---|
| | | Had depression (129) | Had no depression (286) | | |
| Sex of participant | Male | 34(22.5%) | 117(77.5%) | 1 | 1 |
| | Female | 95(36%) | 169(64%) | 1.93 (1.22, 3.06) | 1.63 (0.96, 2.75) |
| Marital status of participants | In Union | 62(24.5%) | 191(75.5%) | 1 | 1 |
| | Not in Union | 67(41.4%) | 95(58.6%) | 2.17(1.42, 3.32) | **2.86(1.66,4.94)** * |
| Educational status of participants | Diploma | 34(41%) | 49(59%) | 2.1 (2.18,1.04) | 1.11 (0.43,2.85) |
| | Degree | 81(29.6%) | 193(70.4%) | 1.34 (0.69, 2.54) | 0.62 (0.28,1.39) |
| | Masters and above | 14(24.1%) | 44(75.9%) | 1 | 1 |
| Monthly income | ≤6192 | 40(26.5%) | 111(73.5%) | 1.6 (0.71, 3.55) | 0.82 (0.32, 2.05) |
| | 6193–8016 | 33(38.4%) | 53(61.6%) | 2.77 (1.19, 6.43) | 2.14 (0.81,5.63) |
| | 8017–10,169 | 47(36.4%) | 82(63.6%) | 2.55 (1.14, 5.71) | 1.50 (0.59, 3.76) |
| | ≥10,170 | 9(18.4%) | 40(81.6%) | 1 | 1 |
| Social support | Poor | 59(34.3%) | 113(65.7%) | 0.73 (0.38, 1.41) | 0.83 (0.39, 1.80) |
| | Moderate | 50(25.6%) | 145(74.4%) | 0.48 (0.25, 0.93) | 0.62 (0.28, 1.35) |
| | Strong | 20(41.7%) | 28(58.3%) | 1 | 1 |
| Work hours per week | ≤40Hrs | 45(25.1%) | 134(74.9%) | 1 | 1 |
| | >40Hrs | 84(35.6%) | 152(64.4%) | 1.65 (1.07, 2.53) | **2.21 (1.23, 3.75)** * |
| Current substance uses | No | 59(24.8%) | 179(75.2%) | 1 | 1 |
| | Yes | 70(39.5) | 107(60.5%) | 1.98 (1.30, 3.02) | **2.73 (1.64, 4.56)** * |
| Job satisfaction | Satisfied | 31(18.9%) | 133(81.1%) | 1 | 1 |
| | Not satisfied | 98(39%) | 153 (61%) | 2.75 (1.73, 4.34) | **3.52 (2.05, 6.07)** * |
| Burnout | No | 64(22.9%) | 216(77.1%) | 1 | 1 |
| | Yes | 65(48.1%) | 70(51.9%) | 3.13 (2.02, 4.86) | **5.11 (2.95, 8.83)** * |

**Foote note**: Hrs.- Hours

ruled by similar civil servant rules. Similarly, our result is consistent with the finding done in Australia [25] which was 32.4%. This might be previous study have included all health professionals with limited work load that might under estimate this problem when we compare with Obstetrics care provider.

However, the finding was lower than the study conducted among health extension workers in rural and urban areas of Ethiopia, which was 43.1% and 36.7%, respectively [26]. This discrepancy might be due to the former study was conducted among health extension workers in which due to the fact that health extension workers are serving different community at their residences and challenged by various attitude of the community, which might increase the problem. In addition, the education opportunity is not available within a short period of time that might worsen the problem.

Similarly, the finding is lower than the finding from a systematic review and meta-analysis conducted in Ethiopia, which was 40% [27]. The discrepancy between a systematic review and meta-analysis conducted in Ethiopia might be due to studies conducted during the COVID-19 pandemic, which might increase the problem as there was a lower understanding of COVID-19 at the beginning.

However, the finding was lower than the studies conducted in Saudi Arabia [28], China [29], Hong Kong [30], Baghdad [31] and Egypt [32] which were 43.9%, 38%, 35.8%, 70.25%, and 71.4%, respectively. The possible justification might be the difference in participants who had different socio-economic and demographic characteristics in the populations as well, and the tools to assess the problem are varied.

On the other hand, the findings of our study are higher than those of the studies conducted in Vietnam [33], and Malaysia [34] which was 13.2%, and 10.7% respectively. In addition, the finding is also higher than the studies in Nigeria 10.7%, and 17.3% [35, 36]. The discrepancy might be due to differences in study participants, as our study only focused on obstetrics care providers, as the obstetrics ward is a stressful environment for them, which might worsen the problem. Similarly, the study conducted in Vietnam [33], and Malaysia [34] has included all healthcare providers, which might include healthcare providers from less stressful wards that might underestimate the problem.

The findings of the study showed that marital status is a significantly associated factor with depression, which is supported by the studies conducted in Dessie Comprehensive Specialized Hospital, Ethiopia [14], and systematic review conducted in Ethiopia [27], Egypt [32], Nigeria [35], Saudi Arabia [28], and Baghdad [31]. This might be because every human being in nature needs love and belongingness, so having a soul mate might prevent loneliness, which might worsen the problem. In addition, living with a spouse or partner was strongly associated with reduced odds of depression which supports the finding [37].

Working hours per week are identified as a significantly associated factor. This is supported by the studies conducted in Saudi Arabia [38], and China [29]. The possible reason might be that working overtime leads to increased work-related stress and other physical disabilities that, in the long term, have significant negative effects on mental health, specifically the chance of developing depression. In addition, working extra limits an individual's free time for relaxation, leading to exhaustion and poor sleep, a combination that might lead to depression. On the other hand, having enough time is crucial for better relaxation, which boosts mood, reduces stress, and improves memory and focus. These benefits enable him or her to return to work with a renewed mindset.

Similarly, substance use is a significantly associated factor in this study setting. This is supported by studies conducted in Ethiopia [14], and Hong Kong [30]. This might mean that substance use may lead to mental health problems during the withdrawal period. This is strengthened by the finding that revealed substance use can trigger or intensify the feelings of loneliness, sadness, and hopelessness often associated with depressive symptoms among participants [39].

This study also identifies job satisfaction as a factor significantly associated with depression. This is in agreement with the study conducted in Southeastern Anatolia from Diyarbakir, Turkey [4]. Job satisfaction is positively correlated with collective efficacy, non-technical skills related to communication and decision-making, and a sense of belonging to their profession [40].

Lastly, burnout is a factor that is significantly associated with depression. This is supported by the studies conducted in Southeastern Anatolia from Diyarbakir, Turkey [4] and in Macao, China [41]. This might be due to a person who has burnout feeling as though "it's never enough" and then might start to feel hopeless or lack all emotion in his or her job and be prone to depression. This is established by the fact that the link between burnout and poor mental health shows that true workplace burnout has a 57% increased risk of workplace absence and a 180% increased risk of developing depressive disorders [42].

## Limitation

Due to the nature of a cross-sectional study, it is difficult to establish a causal relationship between risk factors and depression.

## Conclusion

This study revealed that there was a high level of depression among the obstetric care providers in the study area. Not being in the union in terms of marital status, working more than eight

hours per day, current substance use, lack of job satisfaction, and burnout were factors significantly associated with depression.

## Recommendations

We recommend health professionals take care of themselves and avoid substance use. Secondly, we recommend concerned stakeholders establish mechanisms to improve job satisfaction and overcome burnout by creating different incentive mechanisms that increase staff income, increase the number of staff, provide different rewards, and conduct regular supervision of their problems. Finally, stakeholders and governments should make evidence-based decisions and ensure proper utilization of financial and human resources at health facilities.

## Supporting information

**S1 Table. Supplementary table of responses to each assessment of the tool on depression and burnout.**
(DOCX)

## Acknowledgments

To begin with, our deepest thanks would be to the Madda Walabu University Sheshemene campus for all their support, and our deepest gratitude would go to the West Arsi Zonal health office, each health facility, and data collectors as well as supervisors who contributed to this work. Similarly, we express our heartfelt thanks to the study participants for their time and valuable information.

## Author Contributions

**Conceptualization:** Solomon Seyife Alemu, Mohammedamin Hajure Jarso, Zakir Abdu Adem, Gebremeskel Mulatu Tesfaye, Yadeta Alemayehu Workneh, Wubishet Gezimu, Mustefa Adem Hussen, Aman Dule Gemeda, Sheleme Mengistu Teferi, Lema Fikadu Wedajo.

**Data curation:** Solomon Seyife Alemu, Mohammedamin Hajure Jarso, Zakir Abdu Adem, Gebremeskel Mulatu Tesfaye, Yadeta Alemayehu Workneh, Wubishet Gezimu, Mustefa Adem Hussen, Aman Dule Gemeda, Sheleme Mengistu Teferi, Lema Fikadu Wedajo.

**Formal analysis:** Solomon Seyife Alemu, Mohammedamin Hajure Jarso, Zakir Abdu Adem, Gebremeskel Mulatu Tesfaye, Yadeta Alemayehu Workneh, Wubishet Gezimu, Mustefa Adem Hussen, Aman Dule Gemeda, Sheleme Mengistu Teferi, Lema Fikadu Wedajo.

**Funding acquisition:** Solomon Seyife Alemu, Mohammedamin Hajure Jarso, Zakir Abdu Adem, Gebremeskel Mulatu Tesfaye, Yadeta Alemayehu Workneh, Wubishet Gezimu, Mustefa Adem Hussen, Aman Dule Gemeda, Sheleme Mengistu Teferi, Lema Fikadu Wedajo.

**Investigation:** Solomon Seyife Alemu, Mohammedamin Hajure Jarso, Zakir Abdu Adem, Gebremeskel Mulatu Tesfaye, Yadeta Alemayehu Workneh, Wubishet Gezimu, Mustefa Adem Hussen, Aman Dule Gemeda, Sheleme Mengistu Teferi, Lema Fikadu Wedajo.

**Methodology:** Solomon Seyife Alemu, Mohammedamin Hajure Jarso, Zakir Abdu Adem, Gebremeskel Mulatu Tesfaye, Yadeta Alemayehu Workneh, Wubishet Gezimu, Mustefa Adem Hussen, Aman Dule Gemeda, Sheleme Mengistu Teferi, Lema Fikadu Wedajo.

**Project administration:** Solomon Seyife Alemu, Mohammedamin Hajure Jarso, Zakir Abdu Adem, Gebremeskel Mulatu Tesfaye, Yadeta Alemayehu Workneh, Wubishet Gezimu, Mustefa Adem Hussen, Aman Dule Gemeda, Sheleme Mengistu Teferi, Lema Fikadu Wedajo.

**Resources:** Solomon Seyife Alemu, Mohammedamin Hajure Jarso, Zakir Abdu Adem, Gebremeskel Mulatu Tesfaye, Yadeta Alemayehu Workneh, Wubishet Gezimu, Mustefa Adem Hussen, Aman Dule Gemeda, Sheleme Mengistu Teferi, Lema Fikadu Wedajo.

**Software:** Solomon Seyife Alemu, Mohammedamin Hajure Jarso, Zakir Abdu Adem, Gebremeskel Mulatu Tesfaye, Yadeta Alemayehu Workneh, Wubishet Gezimu, Mustefa Adem Hussen, Aman Dule Gemeda, Sheleme Mengistu Teferi, Lema Fikadu Wedajo.

**Supervision:** Solomon Seyife Alemu, Mohammedamin Hajure Jarso, Zakir Abdu Adem, Gebremeskel Mulatu Tesfaye, Yadeta Alemayehu Workneh, Wubishet Gezimu, Mustefa Adem Hussen, Aman Dule Gemeda, Sheleme Mengistu Teferi, Lema Fikadu Wedajo.

**Validation:** Solomon Seyife Alemu, Mohammedamin Hajure Jarso, Zakir Abdu Adem, Gebremeskel Mulatu Tesfaye, Yadeta Alemayehu Workneh, Wubishet Gezimu, Mustefa Adem Hussen, Aman Dule Gemeda, Sheleme Mengistu Teferi, Lema Fikadu Wedajo.

**Visualization:** Solomon Seyife Alemu, Mohammedamin Hajure Jarso, Zakir Abdu Adem, Gebremeskel Mulatu Tesfaye, Yadeta Alemayehu Workneh, Wubishet Gezimu, Mustefa Adem Hussen, Aman Dule Gemeda, Sheleme Mengistu Teferi, Lema Fikadu Wedajo.

**Writing – original draft:** Solomon Seyife Alemu, Mohammedamin Hajure Jarso, Zakir Abdu Adem, Gebremeskel Mulatu Tesfaye, Yadeta Alemayehu Workneh, Wubishet Gezimu, Mustefa Adem Hussen, Aman Dule Gemeda, Sheleme Mengistu Teferi, Lema Fikadu Wedajo.

**Writing – review & editing:** Solomon Seyife Alemu, Mohammedamin Hajure Jarso, Zakir Abdu Adem, Gebremeskel Mulatu Tesfaye, Yadeta Alemayehu Workneh, Wubishet Gezimu, Mustefa Adem Hussen, Aman Dule Gemeda, Sheleme Mengistu Teferi, Lema Fikadu Wedajo.

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
