## [Decision Letter · Decision Letter 0]

7 Feb 2024

PONE-D-23-33519Prevalence of Depression and Associated Factors among Obstetric Care Providers: Cross-sectional studyPLOS ONE

Dear Dr. Alemu,

Thank you for submitting your manuscript to PLOS ONE. After careful consideration, we feel that it has merit but does not fully meet PLOS ONE’s publication criteria as it currently stands. Therefore, we invite you to submit a revised version of the manuscript that addresses the points raised during the review process.

We look forward to receiving your revised manuscript.

Kind regards,

Engidaw Fentahun Enyew, 

Academic Editor

PLOS ONE

3. In the online submission form, you indicated that [This study is from the project that currently undergoing as result its difficult to provide supporting data. However, up on completion of the project every body get the data by contacting the corresponding author at any time.]. 

Additional Editor Comments:

please address all comments line by line accordingly

Reviewers' comments:

Reviewer's Responses to Questions

**Comments to the Author**

1. Is the manuscript technically sound, and do the data support the conclusions?

Reviewer #1: Yes

Reviewer #2: Partly

2. Has the statistical analysis been performed appropriately and rigorously? 

Reviewer #1: No

Reviewer #2: Yes

3. Have the authors made all data underlying the findings in their manuscript fully available?

Reviewer #1: Yes

Reviewer #2: Yes

4. Is the manuscript presented in an intelligible fashion and written in standard English?

Reviewer #1: Yes

Reviewer #2: Yes

5. Review Comments to the Author

Reviewer #1: Ms. ID 23-33519 Prevalence of Depression and Associated Factors among Obstetric Care Providers:

Cross-sectional study by Solomon Seyife et al.

The aim of this study from thirty-seven health facilities in Ethiopia was to determine the prevalence of depression and associated factors among obstetric care providers working in public health facilities..

General comments

For one month, June 1–30, 2023, a systematic random sampling technique was used to include 423 study participants among obstetrics care providers working in public health facilities. Data were collected through an interviewer-administered questionnaire. The conclusion was that the prevalence of depression among obstetric care providers was 31.1% and was associated to marital status not in union, working more than 40 hours per week, current substance use not being satisfied with their job, and having burnout symptoms. It is an important report from Ethiopia. The line number should be inserted (referred to as missing), so how can I comment on it? Please insert the line number for writing the comment line by line and resubmit again.

Reviewer #2: Review Report

Title: Prevalence of Depression and Associated Factors among Obstetric Care Providers: Cross-sectional study

Manuscript Number: PONE-D-23-33519

Review Comments

The manuscript didn’t abide with the submission guideline of the PONE journal. E.g., the abstract section.

The aim of the study and the scope of the recommendations are different.

Lack of consistency through out the document i.e. Is that depression or major depressive disorder?

The background needs critical revision and the authors should revisit how to write an introduction that convince the reader containing all essential element and entailed flawed sentences E.g. “…by the 2030 year” in the introduction section.

The methods section lacks concise and details. In addition, the sampling technique is doubtable since the obstetric care providers have various levels E.g., Midwifery and Medical doctor. Likewise, the way the dependent variable is described, it fails to differentiate the use either ‘validated’ or ‘structured questionnaire’, this was witnessed by reference number 23. What was the type of the analysis since it is composite variable?

The results and the discussion are relatively better. But both should be presented concisely and completely in logical way. The discussion and the explanations are sometimes against evidence. E.g., HEWs demoralized.

Individual roles of the authors should be explained independently and in detail.

Poor referencing E.g., Reference number 5.

Regards,

6. PLOS authors have the option to publish the peer review history of their article (what does this mean?). If published, this will include your full peer review and any attached files.

Reviewer #1: **Yes: **Asnake Tadesse Abate

Reviewer #2: No

---

## [Author Response · Author response to Decision Letter 0]

1 Mar 2024

Response to Reviewers and Academic Editors

First of all, we sincerely appreciate your contribution to review of our work and your helpful suggestions. Based on AE and reviewers comments and suggestions we try to address each point as following: We also incorporated in revised document with track change and highlights. 

Response to comments and suggestions raised by Academic Editor

Response:

 Regarding the PLOS One journal guideline issue, we corrected the revised document in accordance with the guideline.

 Regarding issues of data availability; previously we have stated that the current manuscript is a part of undergoing research project owned by Madda Walabu University Shashemenne Campus. All relevant data are included within the paper. The data would be guarded carefully by our research team for the only purpose of this scientific study and it is an ongoing project. For this reason and following the indicators of the research review committee of college of health sciences, Madda Walabu University, the authors must not upload the dataset to a stable, public repository. Interested, qualified researchers can access the data by requesting the following concerned bodies. 

The research coordinators of Madda Walabu University Shashemenne Campus Mr. Habtamu Jarso can be contacted

1. Habtemu Jarso (BSc, MPH-E, Assis. Professor)

Shashemene Campus, Madda Walabu University

Institutional email: habtemu.jarso@mwu.edu.et

2. Corresponding author; Solomon Seyife: email: SoleSeifa@gmail.com

Beside this all necessary files have been uploaded on initial submission on the track accordingly. 

 The ethical statement's location was changed in accordance with your suggestion.

2. Response to comments raised by first reviewer 

 Please insert the line number for writing the comment line by line and resubmit again?

Response:

We appreciate your feedback and have already incorporated it into the updated document.

 3. Response to comments raised by second reviewer

1. The manuscript didn’t abide with the submission guideline of the PONE journal. E.g., the abstract section. 

Response:

Thank you very much for your suggestion; On the basis of your suggestion, a correction has been made to the updated document.

2. The aim of the study and the scope of the recommendations are different.

Response:

Thank you very much for your suggestion. Since our study is a single study, it is not advisable to recommend policy makers; instead, it should be in line with our findings. We therefore made a correction after reevaluating our findings.

‘’ We recommend health professionals take care of themselves from substance use, just to live as exemplary for communities. Secondly, we recommend concerned stakeholders to establish mechanisms to improve job satisfaction and overcome burnout by creating different incentive mechanisms that increase staff income, increase the number of staff, provide different rewards, and conduct regular supervision of their problems. Finally, stakeholders and governments should make evidence-based decisions and ensure proper utilization of financial and human resources at health facilities.”

3. Lack of consistency throughout the document i.e. is that depression or major depressive disorder?

Response:

our study is focused on depression. So, we carefully revised the document and made correction in revised document. 

4. The background needs critical revision and the authors should revisit how to write an introduction that convince the reader containing all essential element and entailed flawed sentences E.g. “…by the 2030 year” in the introduction section.

Response:

Thank you very much for your suggestion after revising this section we try to rewrite it for instance based on your recommendation we corrected line number 113. 

 ‘’By 2030, depression symptoms are predicted to overtake other causes as the second largest contributor to disability-adjusted life years (DALYs).’’

5. The methods section lacks concise and details. In addition, the sampling technique is doubtable since the obstetric care providers have various levels E.g., Midwifery and Medical doctor. Likewise, the way the dependent variable is described, it fails to differentiate the use either ‘validated’ or ‘structured questionnaire’, this was witnessed by reference number 23. What was the type of the analysis since it is composite variable?

Response:

Issue of methodology 

Having carefully reviewed the document, we made the necessary corrections. 

 With respect to the sampling method, refer to lines 156–172, as we modified the detailed sampling procedure based on your recommendations. 

‘’The total number of healthcare providers working at an obstetric unit in selected public health facilities, which was obtained from the zonal health bureau, was 708. From similar data obtained from zonal health office those obstetric care providers that serving selected public health facilities were categorized as 455 Midwifery professional, 108 Nursing, 13 IESO, 39 General practitioner, 89 Health officer and 2 Obstetrician. Before the selection of study participants, proportionate allocations to each health facility were made. Finally, from the list, the required sample sizes of 423 obstetric care providers were selected by simple random sampling techniques’’. For further check the document.

 Issue ‘dependent variable’ it fails to differentiate the use either ‘validated’ or ‘structured questionnaire’, this was witnessed by reference number 23

Response:

We have written explicitly about the tool, in particular the dependent variable, validity, and translation of questionnaires. Could you please revise line number 203-216 once? As you pointed out in reference number 23, we only provide the operational definition of the dependent variable; however, we make clear that about the tool in the Data Collection Tool and Procedure section.

 What was the type of the analysis since it is composite variable?

Response:

Based on the operational definitions of each variable we dichotomized those variables and proceed binary logistic regression. For instance for dependent variable. After adding each item score that used to assess depression. The total score of nine items ranging from 0 to 27. We dichotomized as score ≥ 10 were considered to have depression symptoms, and a code ‘1’ was given. While those who scored < 9 were considered to have no depression symptoms and a code‘0’. Other was also in similar fashion… then we run binary logistic regression and multivariable regression accordingly…….some modification was made please refer the revised document. 

6. The results and the discussion are relatively better. But both should be presented concisely and completely in logical way. The discussion and the explanations are sometimes against evidence. E.g., HEWs demoralized.

Response:

Thank you very much for such constructive suggestion we made revision accordingly. 

7. Individual roles of the authors should be explained independently and in detail

Response:

We corrected according to the guide line of PLOS one

‘’ SSA, MH, ZA, GMT, YAW, WG, MAH, ADG, ShMT and LFW have contributed equally to the proposal development, development of the tool, data collection process, and analysis. All the authors have revised the final version of the manuscript and given their approval for publication.’’

8. Poor referencing E.g., Reference number 5.

Response:

Frankly speaking we use some of references direct from web site of different organization that is why some references have lack of full information. However, as much as possible after revision we made a correction accordingly. For instance based on your suggestion reference number 5; we add its URL; Mind Vw. Mood Swings: Causes, Risk Factors, and Ways to Cope. Dotdash Media. 2023. (https://www.verywellmind.com/what-are-mood-swings-1067178)

Concerns about this paper were addressed as follow:

1. Title

A. Why you didn’t include private health facilities (I need justification)

Response:

Due to the fact that most private health facilities do not offer all obstetric and gynecological procedures, we only included governmental facilities. For example, labor and delivery service. Another reason is that the majority of healthcare providers who work in public healthcare facilities are also doing so in the private sector. Finally, the researcher's area of interest is depression among obstetric care providers who work in government health facilities, which have different challenges than private health institutions in terms of work load, income, and other factors.

B. Why you include all obstetric care providers (it is too many) and I need justification.

Response:

We encompass all obstetric care providers because we have to deal with the working environment in which they carry a comparatively similar burden. Furthermore, depression affects everyone equally and is not selective.

2. Current substance use Do you think that this response is correct? Do you think that health care provider respond to this question in our setting???And if so what types of substance they have been used? Try to make it specify, because there are different types of substance

Response:

We appreciate you raising these questions. Though social desirability bias plays a part in the issue, all relevant information had been given prior to data collection, so we hope that they were fully provided with information in accordance. 

Actually, during the process of collecting data our questionnaires, addressed commonly used substance such as alcohol, cigarettes, and Kchat. 

3. Having burnout symptoms (how did you measure it???)

Please refer line number 193-197

A burnout symptom was measured by previously validated tool in Ethiopia. 

Response:

 ‘’Burnout: There are 19 items to assess symptoms of burnout syndrome (19 items) that are categorized into four Likert scales 0 (never), 1 (sometimes), 2 (often), and 3 (always). The sums of the scores of the rating of the items were calculated with a total minimum score of 0 to a maximum score of Healthcare. Obstetrics care providers that scored more than 23 out of 57 were considered to have burnout symptoms (4).’’ 

4. Methology

A. It is better if you mention the name of hospital

Your study population is too many (midwives, gynecologist, nurse, integrated emergency surgical officers, HO) so why you include all of them??? It is better if you add the number of proportionally allocated individuals from each hospital 

Response:

Based on your recommendations we try to narrate accordingly on revised document. For schematic presentation due to many (37) health facilities it’s difficult to put it in one diagram. So to handle this issue we try to explain in narrative form and just it’s self-explanatory; please refer sampling technique procedure. If must we will put it…

B. Obstetrics care providers who have worked for at least six months were included? What is your justification?

Response:

We exclude those health providers due to finally we conclude this study for that study setting. Secondly, those health care providers were not fully engaged in working environments. 

5. Study variable Occupation? Is not necessary. They are health care provider

Response:

Thanks for your correction we accept it and already removed. 

6. Data collection tools

You used English version of quastionaries.so Do you think that all of your study participants equally understand English language, particularly in our setup???

Response:

Thanks for your suggestion but we clearly stated in manuscript as following: please check line number 212-214 

‘’The questionnaire was prepared in English and later translated into Afan Oromo to check the consistence of questionnaire, and English language was used for data collection’’

7. Data quality control

Your data quality control is not detail explained?

Response:

We can add some important points. Refer from revised document 

8. Recommendation

Your recommendation need revision

Response:

Thank you very much for your suggestion. Since our study is a single study, it is not advisable to recommend policy makers; instead, it should be in line with our findings. We therefore made a correction after reevaluating our findings.

---

## [Decision Letter · Decision Letter 1]

30 Apr 2024

PONE-D-23-33519R1Prevalence of Depression and Associated Factors among Obstetric Care Providers: Cross-sectional study PLOS ONE

Dear Dr. Alemu,

Thank you for submitting your manuscript to PLOS ONE. After careful consideration, we feel that it has merit but does not fully meet PLOS ONE’s publication criteria as it currently stands. Therefore, we invite you to submit a revised version of the manuscript that addresses the points raised during the review process.

**ACADEMIC EDITOR: **please address the reviewers comments one by one accordingly. 

We look forward to receiving your revised manuscript.

Kind regards,

Engidaw Fentahun Enyew

Academic Editor

PLOS ONE

Journal Requirements:

Reviewers' comments:

Reviewer's Responses to Questions

**Comments to the Author**

1. If the authors have adequately addressed your comments raised in a previous round of review and you feel that this manuscript is now acceptable for publication, you may indicate that here to bypass the “Comments to the Author” section, enter your conflict of interest statement in the “Confidential to Editor” section, and submit your "Accept" recommendation.

Reviewer #3: All comments have been addressed

Reviewer #4: All comments have been addressed

Reviewer #5: (No Response)

2. Is the manuscript technically sound, and do the data support the conclusions?

Reviewer #3: Partly

Reviewer #4: Yes

Reviewer #5: Yes

3. Has the statistical analysis been performed appropriately and rigorously? 

Reviewer #3: Yes

Reviewer #4: Yes

Reviewer #5: Yes

4. Have the authors made all data underlying the findings in their manuscript fully available?

Reviewer #3: Yes

Reviewer #4: No

Reviewer #5: Yes

5. Is the manuscript presented in an intelligible fashion and written in standard English?

Reviewer #3: No

Reviewer #4: Yes

Reviewer #5: Yes

6. Review Comments to the Author

Reviewer #3: Try to abide the comments given before. conclusion and recommendation section should be revised.

Some grammatical errors should be corrected

Reviewer #4: The questions raised by the previous reviewer has been answered accordingly. I double checked the referenced lines and they look good. I have noted the following:

1. There is not clear justification for the choice of the variable tested for religion and ethnicity. How do they contribute to the depression? Authors should explain to the reader how/why these variables are important. This should be clearly outlined. For example the salary can have an impact on depression duel to the global economic challenges we are witnessing today.

2. Can the authors explain the impact of the different seen in the frequencies for example in the “Types of health providers”. In that section we only have 2 obstetricians and 311 midwifery. Statistically there will be difference and that this mean that obstetricians are not depressed? The appear to be a problem in the frequencies and this has a chance of obscuring the “true” in the analysis and interpretation.

3. In line 356 in the revised manuscript, the number from Vietnam are missing….this information should be provided and the source cited.

4. In line 360 of the revised manuscript, the authors should cite the studies conducted in this countries.

5. From line 370-375; the authors discuss the effect of working hours and those working over 40 hrs a week appear to be more depressed. What did they observe in the individuals working <40 hrs a week? That need to be articulated in the discussion.

6. Response: “We recommend health professionals take care of themselves from substance use, just to live as exemplary for communities.” Should read: We recommend health professionals take care of themselves and avoid substance use”

7. The data used in this study should be anonymized and made available in the public spaces such as FigShare before the work is published.

Reviewer #5: I thank the authors for addressing the reviewer comments. However, i have just a few minor issues/suggestions i want them to address that i believe will be very beneficial to the reader.

Here are the comments:

I suggest the authors include the setting in the title

Example: Prevalence of Depression and Associated Factors among Obstetric Care Providers at public health facilities in the West Arsi Zone, Ethiopia: Cross-sectional study

I request the authors to add a supplementary table with the individual responses (frequencies and percentages) to each of the items in the following: The Patient Health Questionnaire-9, burnout 19 items, and The OSSS-3 sum score.

7. PLOS authors have the option to publish the peer review history of their article (what does this mean?). If published, this will include your full peer review and any attached files.

Reviewer #3: No

Reviewer #4: No

Reviewer #5: No

---

## [Author Response · Author response to Decision Letter 1]

13 May 2024

Response to Reviewers and Academic Editors

First of all, we sincerely appreciate your contribution to review of our work and your helpful suggestions. Secondly we appreciate AE and all reviewers for re-revised our document and put your constructive suggestions and corrections. Based on AE and reviewers comments and suggestions we try to address each point as following: We also incorporated in revised document with track change and highlights. We prepared revised documents in both track change and clear documents based on the journal guide lines. 

Journal Requirements:

 Response: 

We really appreciate your suggestions. Based on your recommendations, we thoroughly checked all references, and we didn’t find any retracted references. We also kindly request that if you get any retracted references, indicate the reference, and we are ready to make corrections at any time during this review period. Thank you very much again.

Response to reviewer -3 

 Try to abide the comments given before. Conclusion and recommendation section should be revised. Some grammatical errors should be corrected

Response:

Thank you very much for your constructive suggestion. We take note of the feedback and have included it into the updated document. 

Response to reviewer -4

1. There is not clear justification for the choice of the variable tested for religion and ethnicity. How do they contribute to the depression? Authors should explain to the reader how/why these variables are important. This should be clearly outlined. For example the salary can have an impact on depression duel to the global economic challenges we are witnessing today.

Response:

We sincerely appreciate your kind inquiry. You provide a clear explanation of the factors of ethnicity and religion. We also recognize that it can be challenging to link those factors to various illnesses, complications, and other health issues. We only included the descriptive portion and did not perform any association analysis with depression for those variables. We included those variables in the descriptive table solely to describe the sociodemographic characteristics of the population under study. 

2. Can the authors explain the impact of the different seen in the frequencies for example in the “Types of health providers”. In that section we only have 2 obstetricians and 311 midwifery. Statistically there will be difference and that this mean that obstetricians are not depressed? The appear to be a problem in the frequencies and this has a chance of obscuring the “true” in the analysis and interpretation.

Response:

Thank you very much for your question. From the beginning, our study was among obstetric care providers, meaning all types of health care providers who serve in obstetrics and gynecology wards. Initially, we obtain the numbers and types of health care providers that serve Obs/Gyn in selected institutions from the zonal health office. 2 obstetricians and 311 midwives were selected as health professionals during the selection process by simple random sampling techniques from all obstetric care providers found in selected institutions.

3. In line 356 in the revised manuscript, the number from Vietnam are missing….this information should be provided and the source cited.

Response:

Thank you very much. We take note of the feedback and have included it into the updated document. 

4. In line 360 of the revised manuscript, the authors should cite the studies conducted in this countries.

 Response:

Thank you very much. We take note of the feedback and have included it into the updated document. 

5. From line 370-375; the authors discuss the effect of working hours and those working over 40 hrs a week appear to be more depressed. What did they observe in the individuals working <40 hrs a week? That need to be articulated in the discussion.

Response: Thank you very much for your suggestions. We accepted and incorporated it in the revised document.

‘’In addition, working extra limits an individual's free time for relaxation, leading to exhaustion and poor sleep, a combination that might lead to depression. On the other hand, having enough time is crucial for better relaxation, which boosts mood, reduces stress, and improves memory and focus. These benefits enable him or her to return to work with a renewed mindset.’’

6. Response: “We recommend health professionals take care of themselves from substance use, just to live as exemplary for communities.” Should read: We recommend health professionals take care of themselves and avoid substance use”

Response:

Thank you very much for your suggestions. We accepted and incorporated it in the revised document.

7. The data used in this study should be anonymized and made available in the public spaces such as FigShare before the work is published. 

Response: Thank very much for your suggestions.

Availability of data and materials

 This current manuscript is a part of undergoing research project owned by Madda Walabu University Shashemenne Campus. All relevant data are included within the paper. The data would be guarded carefully by our research team for the only purpose of this scientific study and it is an ongoing project. For this reason and following the indicators of the research review committee of college of health sciences, Madda Walabu University, the authors must not upload the dataset to a stable, public repository. Interested, qualified researchers can access the data by requesting the following concerned bodies. 

The research coordinators of Madda Walabu University Shashemenne Campus Mr. Habtamu Jarso can be contacted

1. Habtemu Jarso (BSc, MPH-E, Assis. Professor)

Shashemene Campus, Madda Walabu University

Institutional email: habtemu.jarso@mwu.edu.et

Corresponding author; Solomon Seyife: Email: SoleSeifa@gmail.com

Response to reviewer -5

1. I suggest the authors include the setting in the title

Example: Prevalence of Depression and Associated Factors among Obstetric Care Providers at public health facilities in the West Arsi Zone, Ethiopia: Cross-sectional study

Response: Thank you very much for your constructive suggestion. We accepted your recommendations and incorporated in to revised document. 

2. I request the authors to add a supplementary table with the individual responses (frequencies and percentages) to each of the items in the following: The Patient Health Questionnaire-9, burnout 19 items, and The OSSS-3 sum score.

Response: 

We really appreciate your request. We incorporated your request as supplementary tables. You can check it out. We provide the title “Supplementary Table.”

 Thank you very much again for the scientific suggestion.

---

## [Editor Report · Decision Letter 2]

21 May 2024

Prevalence of Depression and Associated Factors among Obstetric Care Providers: Cross-sectional study

PONE-D-23-33519R2

Dear Mr Solomon 

We’re pleased to inform you that your manuscript has been judged scientifically suitable for publication and will be formally accepted for publication once it meets all outstanding technical requirements.

Kind regards,

Engidaw Fentahun Enyew

Academic Editor

PLOS ONE
---

## [Editor Report · Acceptance letter]

29 May 2024

PONE-D-23-33519R2 

PLOS ONE

Dear Dr. Alemu, 

I'm pleased to inform you that your manuscript has been deemed suitable for publication in PLOS ONE. Congratulations! Your manuscript is now being handed over to our production team.

Kind regards, 

on behalf of

Mr Engidaw Fentahun Enyew 

Academic Editor

PLOS ONE